# Metabolic Side Effects from Antipsychotic Treatment with Clozapine Linked to Aryl Hydrocarbon Receptor (AhR) Activation

**DOI:** 10.3390/biomedicines12102294

**Published:** 2024-10-10

**Authors:** Karin Fehsel

**Affiliations:** Department of Psychiatry and Psychotherapy, Medical Faculty, Heinrich-Heine-University, Bergische Landstrasse 2, 40629 Duesseldorf, Germany; fehsel@hhu.de

**Keywords:** hyperglycemia, Akt kinase, brain, adipogenesis, interleukin 6, glucose transporter

## Abstract

Metabolic syndrome (MetS) is the most common adverse drug reaction from psychiatric pharmacotherapy. Neuroreceptor blockade by the antipsychotic drug clozapine induces MetS in about 30% of patients. Similar to insulin resistance, clozapine impedes Akt kinase activation, leading to intracellular glucose and glutathione depletion. Additional cystine shortage triggers tryptophan degradation to kynurenine, which is a well-known AhR ligand. Ligand-bound AhR downregulates the intracellular iron pool, thereby increasing the risk of mitochondrial dysfunction. Scavenging iron stabilizes the transcription factor HIF-1, which shifts the metabolism toward transient glycolysis. Furthermore, the AhR inhibits AMPK activation, leading to obesity and liver steatosis. Increasing glucose uptake by AMPK activation prevents dyslipidemia and liver damage and, therefore, reduces the risk of MetS. In line with the in vitro results, feeding experiments with rats revealed a disturbed glucose-/lipid-/iron-metabolism from clozapine treatment with hyperglycemia and hepatic iron deposits in female rats and steatosis and anemia in male animals. Decreased energy expenditure from clozapine treatment seems to be the cause of the fast weight gain in the first weeks of treatment. In patients, this weight gain due to neuroleptic treatment correlates with an improvement in psychotic syndromes and can even be used to anticipate the therapeutic effect of the treatment.

## 1. Introduction

Metabolic syndrome (MetS), which includes obesity, diabetes, hypertension, hyperlipidemia, and fatty-liver disease, affects more than two-thirds of the U.S. population, and the ever-increasing frequency of MetS is still a major challenge for public health care systems worldwide. There is steadily growing knowledge about the signaling pathways associated with MetS. However, our understanding of how these pathways interconnect to coordinate organismal physiology remains limited. The slow and diffuse onset of MetS further delays the diagnosis. This problem can be addressed by finding discrete triggers for the metabolic disturbances that lead to MetS. Figure 1 illustrates the diseases that are involved in the development of MetS. Moreover, this review provides evidence that disturbed cellular glucose metabolism is the main trigger for each of the diseases. Reduced glucose uptake due to insulin resistance, stress, obesity, or massively increased glucose consumption due to viral infection leads to cellular glucose deprivation and decreased energy production. Evolutionary conserved survival pathways are started, which include activation of the transcription factors AhR and HIF-1α. Their activation is linked to iron depletion. That is why all the diseases shown in Figure 1 are accompanied by iron deficiency and anemia [1].

Lipophilic persistent organic pollutants enter the environment by discharging into surface water and diffusing into the air. For a long time, these environmental pollutants have been suspected to contribute to MetS. They act as ligands of the aryl hydrocarbon receptor (AhR). Activation of the AhR influences a variety of physiological and pathological processes involved in insulin secretion and glucose metabolism far beyond just xenobiotic metabolism of chemicals and detoxification in the liver [2]. However, the results are fragmentary and concentrate on liver metabolism. This review highlights the decisive role of the AhR in the coordination of several metabolic pathways in the liver, brain, and adipocytes.

After ligand binding in the cytosol, the AhR dissociates from its inhibitory proteins HSP90 and XAP2, forms a complex with its common partner, AhR nuclear translocator (ARNT), and enters the nucleus. As a transcription factor, it selectively binds to specific sequences known as xenobiotic responsive elements in the promoters of target genes to enforce transcription of specific genes involved in xenobiotic metabolism, such as CYP1A1 or Cyp1B1. Besides this canonical AhR signaling, noncanonical AHR activation influences various signaling pathways such as epidermal growth factor receptor (EGFR), signal transducer and activator of transcription 3 (STAT3), hypoxia-inducible factor-1α (HIF-1α), nuclear factor kappa B (NF-κβ), and sex hormone signaling pathways [2]. Serum levels of AhR ligands are dose-dependently associated with MetS and mitochondrial dysfunction [3]. The AhR is expressed in various tissues, including the pancreas, liver, brain, and adipose tissues, making it a crucial receptor in many diseases [4] (Figure 2). 

The congruency of the diseases linked to MetS (Figure 1) and AhR activation (Figure 2) is striking and predicts a decisive role of the AhR in the development of MetS. However, developing the pathological features of MetS due to environmental pollutants is a lengthy process, requiring longitudinal studies over years or even decades. In contrast, treatment of schizophrenic patients with antipsychotic drugs induces metabolic changes within weeks. This allows for a close monitoring of diverse physiological parameters. Weight gain is observed in 70% of the patients, and half of the patients show metabolic decompensation with transiently elevated liver transaminases. Moreover, animal studies with these drugs provoke MetS, too. In addition, these studies give insights into the pathophysiology of different organs, which leads to MetS. 

Although the structurally related, atypical antipsychotic drugs clozapine and olanzapine have a favorable therapeutic profile in schizophrenia and are virtually devoid of extrapyramidal motor side effects, their endocrine and metabolic side effects induce a variety of metabolic complications. Henderson et al. [17] reported that more than one-third of schizophrenics treated with clozapine for 5 years developed diabetes. Weight gain, obesity, and related metabolic abnormalities such as hyperglycemia and dyslipidemia may evolve into MetS with a high risk for future cardiovascular morbidity and mortality. Prevalence rates of 28 to 46% for MetS in patients taking antipsychotic drugs have been reported [18]. In line with the observations on patients, animal studies have shown hyperlipidemia, raised triglyceride levels, along with high levels of fasting blood glucose, too [19,20]. Similar to exposure to environmental pollutants, clozapine treatment increases immunosuppression, the risk of bone marrow toxicity [21,22], and the incidence of diabetes and abdominal obesity [23,24,25,26] and reduces selenium plasma levels [27] and body iron [28] as observed in rats treated with polyhalogenated hydrocarbons [29]. In 2022, it was shown that clozapine, in contrast to olanzapine, concentration-dependently activates the AhR in the liver, aorta, and preadipocytes with harmful effects on vasodilatation and adipogenesis [30].

Among the antipsychotics, clozapine has the highest risk for the development of MetS [18]. Therefore, this review addresses the metabolic and biochemical changes from clozapine treatment and presents an attempt to integrate the plethora of significant but previously independent puzzle pieces within this framework of MetS, including glucose and iron deficiency, AhR, AMP-activated protein kinase (AMPK), and HIF activation. Understanding the interlocked survival pathways might provide relevant adjusting screws for the development of more effective treatments against MetS or even for its prevention.

## 2. Drug-Induced Metabolic Effects in Patients from Clozapine Treatment

### 2.1. Inflammation

Due to the risk of life-threatening agranulocytosis, clozapine is titrated up slowly with weekly controlled blood counts and other parameters. Significantly elevated levels of CRP, IL6, and other cytokines point to an inflammatory response at the beginning of treatment [31]. IL6 is even higher than the elevated levels found in both peripheral blood and cerebrospinal fluid of patients with schizophrenia and patients with MAFLD compared to healthy controls [18,32]. A significant increase in neutrophils within one week of treatment is considered as an additional non-specific inflammatory marker [33], but given that IL6 inhibits neutrophil apoptosis [34], it cannot be ruled out that the higher number of neutrophils results from prolonged survival instead from increased production.

Under longer treatment with clozapine, the proinflammatory response shifts to an anti-inflammatory response. Serum levels of the anti-inflammatory cytokine IL4 were 83% higher in clozapine-treated patients than in unmedicated controls [35]. Il4 is a key regulator of glucose/lipid metabolism and regulates hypothalamic appetite control [36]. It downregulates agouti-related protein in the hypothalamus of clozapine-treated animals, while olanzapine-induced hyperphagia is based on the increased expression of the orexigenic agouti-related protein [37]. Furthermore, IL4 induces immunosuppression by arginase upregulation in immune cells and increasing the number of regulatory T cells [38]. It even increased plaque clearance by arginase+ microglia in a mouse model of Alzheimer’s disease [39]. Furthermore, activation of the type II IL-4 receptor increases insulin-independent glucose uptake [40,41]. That is probably the reason why IL4 dampens AhR activation [42].

While IL4 is expressed only in T helper cells, IL-6 is produced under stress and especially glucose deprivation [43] by many cell types, including hematopoietic cells, myocytes, endothelial cells, and adipocytes [44]. It instructs hepatic gluconeogenesis in the absence of hypoglycemia. Furthermore, IL6 promotes hepcidin production in the liver [45].

### 2.2. Hepatotoxicity

Clozapine is extensively metabolized in the liver, and a transient asymptomatic elevation of liver transaminases, especially aminotransferases, has been observed in up to 50% of patients [46]. The liver upregulates transamination reactions under clozapine to cope with the metabolic derangement due to greater gluconeogenesis and insulin resistance [47]. Patients with higher clozapine plasma levels and male patients were at a higher risk for liver damage. Lee et al. [48] even suggested the possibility of using serum alanine aminotransferase level as an early indicator for MetS in patients treated with clozapine. Within the first 13 weeks of treatment, most of the increase in the different enzymes remitted—probably with the help of IL6 [46]. 

### 2.3. Hyperglycemia

Type 2 diabetes mellitus (T2DM) is one of the most common side effects of clozapine, and patients with obesity or hypertriglyceridemia have a higher risk for the occurrence of abnormal glucose metabolism, regardless of the type of antipsychotic medication [49]. Glucose metabolism is dysregulated even in the absence of weight gain. Clozapine binds to the M3 receptor on beta cells and reduces insulin secretion [50] (Figure 3). Alternatively, activation of AhR disrupts insulin secretion and glucose homeostasis in an AhR-dependent manner, too [51]. Without insulin, glucose cannot be taken up by cells and remains in the blood. In addition, higher blood glucose levels can result from increased hepatic glucose output. Inhibition of serotonergic signaling via 5-HT1A/5-HT2A receptors shifts hepatic glycogen synthesis toward gluconeogenesis with increased glucose export into the blood [52]. Hopefully, the glucagon-like peptide-1 receptor agonist liraglutide significantly reduces glucometabolic disturbances and body weight in insulin resistant, over-weight patients treated with clozapine or olanzapine [53].

### 2.4. Weight Gain

Nearly 70% of the patients under clozapine suffer from weight gain. Slim patients were more likely to experience significant weight gain of more than 10% of body weight. The significant increase in body mass index is accompanied by increased triglyceride and decreased high-density lipoprotein levels. It results from the imbalance between energy intake and energy expenditure [54]. While energy uptake under clozapine is not increased, energy expenditure is reduced. Brown adipose tissue is a primary site of energy expenditure through thermogenesis. Clozapine inhibits brown adipogenesis [55], leading to decreased thermogenesis and reduced body temperature [56]. In contrast, clozapine favors white adipogenesis in vitro [57]. It increases preadipocyte differentiation as well as lipid droplet formation in mature adipocytes. 

Basal metabolism, the energy utilized at rest, comprises approximately 60% of the total daily energy expenditure. Consistent ingestion despite decreased basal metabolic rate provokes a surplus of calorie intake. The liver converts this excess of nutrients into triglycerides, which are packaged into low-density lipoproteins and transported to the adipose tissue for long-term energy storage [58]. 

### 2.5. Iron Depletion

Only two studies and several case reports describe anemia from clozapine treatment [33,59]. In general, iron metabolism is highly regulated by transcriptional and post-transcriptional mechanisms at both the cellular and systemic levels. Iron balance is maintained by iron acquisition and recycling. Iron is transported from the gut through the duodenal enterocytes into the blood, where it is bound to transferrin [1,60]. Transferrin binds to the transferrin receptor (CD71) on cell membranes. After endocytosis of the iron/transferrin/CD71 complex, iron is liberated from transferrin and bound to ferritin. The free iron pool in the cell is very low to minimize oxidative stress. AhR activation downregulates CD71 expression and, therefore, reduces cellular iron uptake [6]. 

In addition, clozapine inhibits the absorption of dietary iron. An animal study revealed high levels of hepcidin, a hormone produced in the liver and in fat [61]. IL6 induces hepcidin expression, leading to an iron deficit and, in the end, to anemia [1]. Studies on iron absorption revealed a role for AhR in hepcidin and ferritin expression, too [7,62]. Under ER stress, ferritin expression is upregulated, and cellular iron is stored in ferritin cages that contain up to 4,500 atoms of iron/cage [63]. Subsequently, the production of Fe-S- clusters in the mitochondria and cytosol is disrupted [64]. Iron deprivation inhibits mitochondrial aconitase activity and processing of citrate within the Krebs cycle. Citrate is redirected to the cytoplasm, where it is used for fatty acid biosynthesis and lipid droplet formation [65]. During clozapine treatment, the upregulation of the citrate carrier and accumulation of lipid droplets was seen in adipocytes [57], male liver [26], and even in neutrophils [66]. In the liver of the male rats, heme deficiency increased the expression of the heme-regulated inhibitor (HRI)) which forces the recycling of iron from senescent red blood cells [20,61]. Upregulation of hepcidin, together with the anemia of the male clozapine-treated rats and iron deposition in female livers, point to a desired shortage of iron during clozapine medication [61].

### 2.6. Sex Differences

Estrogen and testosterone exert beneficial systemic effects and protect against MetS and T2DM [67]. Loss of circulating sex hormones during menopause induces rapid changes in whole-body metabolism that contribute to MetS as well as osteoporosis and neurological diseases [68,69]. The prevalence of MetS increases with age in both genders, with earlier appearance in men [70]. Thus, estrogen has a strong protective impact on MetS. Its binding to the estrogen receptor (ESR) 1 induces Akt activation and cellular glucose uptake while binding to ESR2 triggers the opposite. Aging is accompanied by a shift from ESR1 to ESR2 expression and reduced Akt activation [69]. In line with this, ESR1 knockdown triggered several features of MetS in mice, such as weight gain, increased visceral adiposity, hyperphagia, and hyperglycemia [71]. 

A few studies have investigated sex-related differences in clozapine’s tolerability, reporting that women experienced more frequent weight gain, hyperglycemia, urinary incontinence, and constipation, while men experienced hypertension and dyslipidemia [72,73,74]. Sex differences were also observed, with neutropenia happening more often in women, while cardiomyopathy and myocarditis happened more often in men [75]. For comparison, secondhand smoke, which activates the AhR, was more closely related to MetS, abdominal obesity, and high blood pressure in men than in women [76].

More profound studies on the sex-dependent side effects of clozapine were conducted with rats. All confirm the main findings in patients except weight gain. While female patients have a higher risk for weight gain, female rats are protected from it [26,77]. This discrepancy is easily resolved by comparing the ages of studied animals and patients. While female rats are young and premenopausal, women with MetS are older and usually postmenopausal. This difference underlines the strong protective effect of estrogen on metabolic diseases. Estrogen signaling via ESR1 controls insulin sensitivity in female mice and is necessary for maintaining glucose homeostasis in both male and female mice [78]. Furthermore, estrogen ameliorates hyperglycemia-induced AMP-dependent protein kinase (AMPK) inactivation [79]. AMPK ensures insulin-independent glucose uptake via the Akt2/Glut4 pathway. Moreover, AMPK extends the time of Glut4 presentation on the cell surface by slowing the recycling of Glut4 [80]. 

**Figure 3 biomedicines-12-02294-f003:**
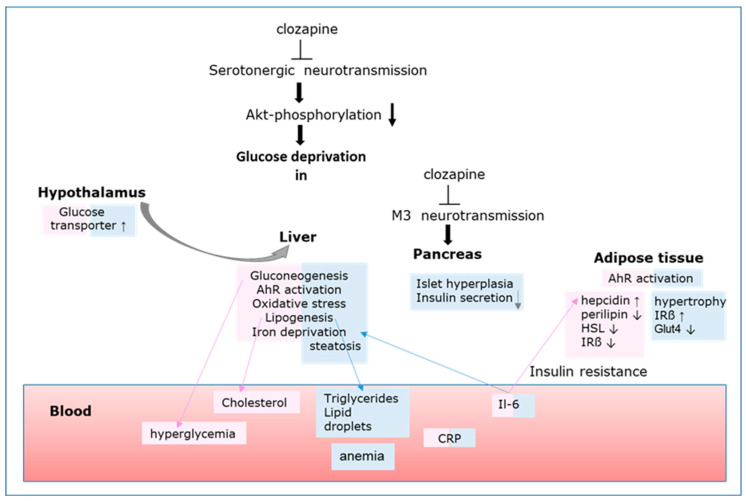
Schematic diagram of the organs’ crosstalk during clozapine treatment [19,20]. Sex-specific changes are highlighted in pink for females and blue for male rats. Arrows indicate up- or downregulation of the respective proteins. It is still an open question whether the brain/hypothalamus orchestrates all metabolic changes or whether neurotransmitter blockade in liver, pancreas, and fat tissue directly leads to the reprogramming of the cell’s metabolism. In the pancreas, clozapine decreases insulin secretion and provokes islet hyperplasia by M3 receptor antagonism [81,82].

Male rats under clozapine treatment show significant weight gain due to an increase in adipose tissue and liver fat [26]. Adipocyte hypertrophy is linked to insulin resistance (Figure 3). Although the insulin receptor is upregulated on adipocytes, the insulin-dependent glucose transporter (Glut) 4 is downregulated. In female rats, perilipin, the major lipid droplet coat protein in mature adipocytes, is downregulated, which facilitates lipolysis by the hormone-sensitive lipase (HSL). However, HSL is downregulated, too (Figure 3). 

Another main difference between the sexes relates to hepatic lipogenesis. Clozapine alters the hepatic expression of cholesterol synthesis genes in an AhR-dependent manner in rodents and humans [83]. Increased SREBP-2 expression in female rats is linked to higher cholesterol production. In particular, the low-density lipoproteins are significantly elevated. They can activate Akt and glucose uptake instead of insulin, thereby mitigating metabolic disturbances [84]. In the male liver, the AhR target SREBP-1C is upregulated, while the AhR triggers proteolytic degradation of mature SREBP-2 [20,83]. SREBP-1C forces triglyceride synthesis and leads to hypertriglyceridemia, fatty liver, and adipocyte hypertrophy. 

Although both sexes show higher fasting glucose levels during clozapine treatment, hyperglycemia is significant only in female rats [26]. It is not attributed to insulin resistance (Figure 3) but to high hepatic glucose output. This is not only a primary effect in the liver but also mediated by the CNS. Neurons in the hypothalamus trigger hepatic glucose output via sympathetic innervation in the liver. Increased glycogenolysis and gluconeogenesis can be stopped either by insulin [85] or by activation of neurons inside the dorsal motor nucleus of the vagus nerve [86]. Both clozapine and olanzapine inhibit the activation of these neurons, and hepatic glucose output is not counterregulated. However, AhR activation suppresses gluconeogenesis and glycogenolysis, stimulates lipogenesis, and triggers inflammatory gene expression in male rodents [20,87].

## 3. Mechanisms of Metabolic Changes Leading to MetS

### 3.1. Clozapine Inhibits Akt Activation and Cellular Glucose Uptake

Clozapine is a pharmacologically ‘dirty’ drug, binding to serotonergic, dopaminergic, muscarinic, adrenergic, and histaminergic receptors [88], and its therapeutic efficiency may even rely on this broad receptor binding profile. Receptor antagonism disturbs the PI3K/Akt pathway that guarantees sufficient glucose import [50,69]. Moreover, increased methylation of the *akt1* and *akt2* genes was found in patients treated with atypical antipsychotics [89]. 

The Akt protein family consists of 3 serine/threonine kinases, which are involved in many physiological processes. Akt1 is expressed ubiquitary, while Akt3 is active in neurons. Akt2 is primarily activated by insulin signaling. The Akt isoforms represent core nodes, where external signals are transformed into intracellular reactions. Phosphorylated Akt inhibits apoptosis, increases cell proliferation, and is beneficial for nitric oxide production and glucose uptake. By inhibition of glycogen synthase kinase (GSK) 3ß, Akt increases glycogen synthesis and energy storage. 

Glucose uptake depends on the abundance of glucose transporters (Gluts) on the cell membrane. Gluts are stored in vesicles beneath the plasma membrane, and pI3K/Akt signaling causes the fusion of these vesicles with the membrane, resulting in the translocation of Gluts into the cell membrane [69]. Diverse receptors for hormones, neurotransmitters, and growth factors activate the PI3K/Akt-triggered glucose uptake, f.e. the hormone insulin [90]. Therefore, lowering hormone production under clozapine or insulin resistance in T2DM reduces cellular glucose uptake. In obese or hyperglycemic subjects, the number of Gluts on the cell surface of blood cells is reduced [91,92]. In line with these findings, neurotransmitter blockade by APDs, as well as glucocorticoids secreted along the hypothalamic–pituitary–adrenal (HPA) axis, inhibit Akt signaling [93,94] and reduce the intracellular glucose level [95]. Glucose deprivation disturbs the glycosylation of proteins in the ER, including Akt [96,97], which leads to a transient stop of translation and the induction of several survival pathways (Table 1). In general, four main mechanisms are proposed: (1) AhR activation; (2) Hypoxia-inducible factor (HIF); (3) ATF4; (4) Inflammation.

### 3.2. Glucose Deprivation Triggers AhR Activation

AhR-activating ligands can derive from various endogenous or exogenous sources. The gut microbiome is a major source of endogenous AHR ligands because several tryptophan-derived compounds are generated through the microbial metabolism of tryptophan [107]. Alternatively, IL6 activates the indoleamine 2,3-dioxygenase 1 (IDO-1), a key enzyme in converting tryptophan into kynurenine, another AhR ligand [108]. In mice, kynurenine treatment causes obesity, liver steatosis, and hyperglycemia despite a low-fat diet [109]. Bilirubin, biliverdin, and modified low-density lipoproteins can activate the AhR in hepatocytes as well. During liver insufficiency, serotonergic dysfunction induces AhR activation and CYP1A1 expression [110]. The same goes for blocking serotonergic signaling by clozapine [30]. In addition, glucose deprivation triggers nuclear translocation of the AhR in HepG2 cells [99] and increases AhR transcription in gliomas [111]. The PAH dioxin significantly attenuates insulin-induced glucose uptake dose-dependently in 3T3 cells. The subsequent glucose deprivation might even be part of the extremely strong AhR response [112,113] and the reason for an additional noncanonical AhR activation by dioxin. Similarly, another PAH, bisphenol A, induces abnormal glucose metabolism and insulin resistance by activating the AhR [114]. 

Omeprazole, doxorubicin, and aflatoxin B1 are called noncanonical AhR ligands [115,116,117]. They trigger AhR activation—probably by reducing glucose uptake via disturbed Akt signaling [118]. AhR activation by clozapine also relies on Akt inactivation and glucose deprivation [67,119,120].

Recently, Swanda et al. [98] uncovered a stress-induced pathway leading to endogenous AhR ligand production. Lysosomal cystine shortage is linked to tryptophan degradation and the synthesis of kynurenine and other potent AhR activators. In patients with obesity and metabolic syndrome, plasma total cysteine is increased [121] due to decreased glutathione synthesis and increased oxidative stress [122]. In schizophrenic patients, under clozapine treatment, high kynurenine, and glutamate plasma levels point to AhR activation and high activity of the cysteine/glutamate transporter xCT in vivo [123,124]. 

Besides the ligand-dependent activation of the AhR, phosphorylation by GSK 3ß is essential for the best activation of AhR-triggered gene transcription [104]. Under physiological conditions, Akt keeps GSK3ß activity in check. Thus, inactive Akt entails dephosphorylation and, thereby, activation of the GSK3ß. It inhibits glucose storage as glycogen when glucose import is downregulated due to missing Akt activation. In addition, GSK3ß triggers NFκB activation with increased proinflammatory IL6 production [125] and phosphorylation of the AhR.

The Cyp1A1, Cyp1A2, and Cyp1B1 genes are well-known AhR targets [126], and clozapine is metabolized either by Cyp1A2 to N-desmethylclozapine or by Cyp3A–not regulated by the AhR- to clozapine-N-oxide [127]. The metabolism is sex-dependent, and the female liver produces more N-desmethylclozapine, while the N-oxide plasma level revealed no sex difference in schizophrenic patients [128]. Interestingly, the clozapine metabolite, N-desmethylclozapine, inhibits glucose uptake in neuronal cells more potently than the parent drug, whereas clozapine N-oxide was essentially inactive [129]. However, the gender differences vanish after 24 weeks [130], which is in line with the findings that continuous exposure to AhR ligands for a longer time reduces CYP1A, Cyp1A2, CYP3A, and AhR transcriptional expressions [131]. Down-regulation of these cytochromes is linked to MetS, hypertension, and hypercholesterolemia in rat models [132,133,134]. 

Cyp1B1 is a further AhR target gene. This cytochrome is an oxidative stress marker inside the mitochondria. At that stage, it converts melatonin to N-acetylserotonin, which is able to activate autophagy and Akt signaling [135,136]. Moreover, brain Cyp1B1 plays critical roles in neuroprotective steroid synthesis [137] and in cerebrovascular and dopamine homeostasis [138]. Cyp1B1 even increases cell tolerance to ferroptosis, a distinct form of programmed cell death that is characterized by an iron-dependent accumulation of lipid peroxides [139].

### 3.3. AhR Triggers ATF4 Activation

ER stress, which is caused either by hypoxia or glucose deprivation, induces the unfolded protein response with activation of the protein kinase R-like ER kinase (PERK) and phosphorylation of the eukaryotic initiation factor (eIF) 2α [140]. Although translation is transiently stopped by peIF2α, the activating transcription factor 4 (ATF4) is upregulated as part of the unfolded protein response along the PERK-eIF2α-ATF4 axis. In addition, ATF4 expression is induced by the AhR [99]. Given that endolysosomal cystine depletion leads to AhR activation, it is not surprising that ATF4 increases the expression of transporters for cysteine and cystine, named alanine serine cysteine transporters (ASCTs) and xCT (Figure 4). Stress and nutrition limitations promote cysteine to antioxidative glutathione [GSH) flux to counter nutrition limitation-induced ROS [141]. Increased cysteine and cystine uptake guarantees sufficient protection against oxidative stress through the synthesis of GSH, glutaredoxins, thioredoxins, and peroxiredoxins. Moreover, cysteine is a semi-essential amino acid. Minor amounts of cysteine can be generated by transsulfuration of methionine and homocysteine. This pathway relies on the activity of cystathionine β-synthase and cystathionine-γ-lyase, which are not only crucial for cysteine production but also for the generation of hydrogen sulfide [142]. In line with increased GSH production by upregulation of ATF4, clozapine was shown to significantly increase ATF4 protein and mRNA levels in HL60 cells and adipocytes [30,66]. Upregulated GSH serum levels were found in clozapine-treated mice and patients [143,144].

### 3.4. AhR Triggers HIF-1 Activation

AhR activation suppresses transferrin receptor (CD71) expression and upregulates intracellular ferritin levels, leading to low intracellular iron levels. Additionally, AhR-induced expression of Cyp1B1 and the transcription factor ATF4 effectively counteract ferroptosis, presumably via increased cysteine production and enhanced antioxidant response [98,147].

AhR-mediated iron depletion under glucose deprivation is part of a survival pathway, which activates the hypoxia-induced factor (HIF) 1. This transcription factor consists of 2 subunits. HIF-1β is a stable, constitutively expressed nucleoprotein, while the HIF-1α subunit has a short half-life because of its degradation by propyl hydroxylases. As the name indicates, hypoxia triggers HIF-1α stabilization by deactivation of the degrading oxygen-dependent enzymes [148]. Under glucose deprivation, increased ferritin expression and reduced iron uptake lead to a withdrawal of iron from the propyl hydroxylases, leading to their deactivation and to the stabilization of HIF-1α. HIF-1 activation induces the genes for Glut1, Glut3, BDNF, and VEGF, with simultaneous inhibition of PTEN expression [149]. It triggers energy production toward aerobic glycolysis, which needs much more glucose for ATP production than oxidative phosphorylation. Ongoing iron depletion leads to mitochondrial dysfunction because electron transport along the electron transport chain depends on the iron-sulfur clusters of the complexes I, II, and III [150]. This mitochondrial dysfunction is a hallmark of metabolic dysregulation and the development of MetS [151]. During clozapine treatment, the F0F1 ATPase inhibitory factor is induced, leading to reduced activity of the respiratory chain. Accordingly, the oxygen consumption of the cells is reduced [66]. 

Iron deficiency is associated with an increase in IL6 and CRP levels [152], and both parameters were elevated at the beginning of clozapine treatment [31,153]. IL6 and Cyp1B1 increase the hepatic production of hepcidin [154], which downregulates the iron uptake in the duodenum. Long-lasting iron depletion affects erythropoiesis, even though erythroblasts represent a privileged destination for circulating iron. Reduced erythropoiesis results in anemia of chronic disease, a common condition in patients with MetS (Figure 1) [1]. In line with these results, male rats fed with clozapine are anemic, while female rats store iron as hemosiderin in their livers [61].

### 3.5. Glucose Deprivation Triggers Proinflammatory Response

In general, glucose deprivation is correlated with increased cytokine levels, especially IL6 [43]. ‘Hungry’ cells secrete IL6, which attracts immune cells. In these cells, IL6 induces the Stat3/HIF-1/VEGF pathway (Figure 2), and VEGF secretion triggers Akt-linked glucose uptake into the glucose-deficient cells [106]. In neuronal cells, IL6-triggered upregulation of gastrin-releasing peptide and its receptor activates PI3K/Akt signaling without the help of immune cells [155].

At the beginning of clozapine treatment, IL6 and CRP levels are elevated [31,153]. Moreover, significantly increased IL6 levels were determined in COVID-19-infected hospitalized patients [156], often linked to hyperglycemia. In depressive and schizophrenic drug-naive patients as well as AD patients, IL6 levels are elevated, too [157,158,159]. In a longitudinal study, CRP and IL6 levels seem to be reliable predictors of MetS incidence and persistence, respectively [160]. Under insulin resistance, stressed adipocytes secrete high levels of IL6, which induce proinflammatory responses in the fat tissue. Downregulation of Glut4 despite increased insulin receptor ß chain expression points to insulin resistance and decreased glucose uptake in male clozapine-treated rats [26]. IL6 activates several signaling pathways, including help from immune cells [106], ferritinophagy [161], and AhR induction [108], which in turn restrains IL6 levels [162] and further immune responses.

### 3.6. Viral Infection Reduces Glucose Availability

We all know that patients with MetS and its comorbidities (Figure 1 magenta circles) have an increased risk of COVID-19 infection and probably have aggravated symptoms [163]. 

It is less known that the COVID-19 pandemic increases the average probability of new-onset MetS by 4.4% in the overall population [164]. Viral infection and proliferation induce hyperglycemia [165] and the reprogramming of host glucose metabolism. Hepcidin, ferritin, CRP, and IL-6 are predictive parameters of critical COVID-19 occurrence [166], and extremely high levels of IL-6 are strongly associated with the presence of septic shock or sepsis in COVID-19 patients [167]. The immune responses are triggered by HIF-1 [168]. This transcription factor increases glucose uptake and shifts metabolism toward aerobic glycolysis, which is necessary for the activation of immune cells as well as viral replication [169]. In line, SARS-CoV-2 replication is inhibited by 2-deoxy-D-glucose, a glycolysis inhibitor [170]. Shin et al. demonstrated that the insulin/insulin receptor/Akt2 pathway was inhibited in various tissues by the SARS-CoV-2 infection [171]. This glucose deprivation in virus-infected cells activates the AhR and promotes replication, thereby increasing lung pathology [172,173]. Massive energy deficits in these cells even activate AMPK, which increases glucose transporter translocation in the lung and heart of SARS-CoV-2-infected cats [172]. 

### 3.7. Clozapine Induces Metabolic Reprogramming in HL-60 Cells [66]

As depicted in Figure 3, neuroreceptor inhibition under clozapine transiently reduces Akt activation and cellular glucose import. Intracellular glucose deprivation induces ER stress [81]. In the subsequent unfolded protein response (UPR) [66] (Figure 4) the activities of several redox-active ER proteins are regulated by S-glutathionylation [174]. This high GSH consumption lowers the cystine levels, which triggers kynurenine synthesis and AhR activation. The AhR reduces intracellular iron, leading to the stabilization of HIF-1 and increased transcription of glucose transporters and glycolytic proteins. The metabolism of the cells shifts toward glycolysis, and the Krebs cycle slows down. Excess mitochondrial citrate is transported back to the cytoplasm, where it is used for lipogenesis and lipid droplet formation. Within 24h of clozapine treatment, the metabolism of HL60 cells is reprogrammed again toward oxidative phosphorylation, but in a reduced mode. Mitochondrial respiration is dampened by upregulation of the F0F1-ATPase inhibitory factor 1, leading to 30–40% lower oxygen consumption but sufficient ATP production in HL60 cells. Downregulation of mitochondrial respiratory functions was previously observed in patients with metabolic syndrome [175]. This cellular adaptive stress response mechanism, called mitohormesis, augments resistance against stressors by generating reactive oxygen species (ROS) [176,177]. Clozapine-induced mitohormesis is, therefore, an excellent way to escape either ATP or glucose deficits and to make the cells more resistant to apoptosis.

### 3.8. Akt Inhibition Entails All Facets of MetS

Clozapine-mediated Akt inhibition directly disturbs vascular homeostasis by inactivating endothelial nitric oxide synthase, thus impairing nitric oxide production and vasodilatation [30,178]. In addition, missing GSK3ß phosphorylation by Akt leads to reduced synthesis of glycogen, a key point in the energy reserve of glucose metabolism. Low glycogen levels are linked to MetS and increased IL6 production [179]. While functional Akt signaling increases expression of xCT, import of cystine, and production of GSH [180], Akt inactivity under clozapine leads to significantly reduced GSH levels in patients and, conversely, higher activity of glutathione peroxidases [181]. Further physiological changes are triggered by AhR activation. It directs adipogenesis under clozapine [30] and downregulates iron metabolism. A consequence of this iron deprivation is iron deposition in the heart and liver. Although no T2* magnetic resonance imaging of the hearts exists, one can speculate that cardiac iron deposition contributes to the QT interval prolongation during clozapine medication [182], as previously described for patients with MetS [183]. Hepatic iron loading, which is detected in female, clozapine-treated rats, alters lipid homeostasis by inducing SREBP-2-mediated cholesterol biosynthesis [184]. In male rats, the AhR is more active and induces SREBP-1C expression, leading to fatty liver, hypertriglyceridemia, and obesity [26]. Moreover, SREBP-1C upregulates hepcidin levels, thereby inhibiting iron uptake in the duodenum and iron liberation from iron stores. In a MASLD model, increased SREBP-1C activity stimulates iron accumulation and fibrosis-related gene expression in mouse hepatic stellate cells [185]. Furthermore, AhR-mediated downregulation of the transferrin receptor, together with a higher ferritin light chain, reduces intracellular iron levels, thereby stabilizing HIF-1. This transcription factor increases transcription of glucose transporters but simultaneously shifts energy production from oxidative phosphorylation to glycolysis. This shift circumvents possible mitochondrial dysfunction due to iron depletion. The crosstalk between glucose/lipid and iron metabolism and the pathways involved in clozapine-triggered MetS are depicted in Figure 5. 

## 4. Animal Models of MetS with Known AhR Participation

A lot of animal studies exist; they try to unravel the metabolic changes leading to MetS. In Table 2 three of them are compared with the different triggers clozapine [26], dioxin-like coplanar 3,3′,4,4′,5-pentachlorobiphenyl (PCB) 126 [186], and alcohol [187]. For better comparison, they were conducted using the same animal model: the Sprague–Dawley rat. However, the time of exposure varied from 26 days to 90 days. A gender difference was considered during treatment with PCB126 and clozapine. 

All three chemicals are known to reduce glucose uptake and to activate the AhR [30,186,187,188]. In turn, the activated AhR affects glucose homeostasis and liver functions (Table 2) through the downregulation of peroxisome proliferator-activated receptors (PPARS) [13]. PPARs are a class of ligand-activated transcription factors, which regulate lipid and carbohydrate metabolism [189]. Accumulating evidence supports a link between the PPARs and diabetes, obesity, dyslipidemia, and inflammation. In adipocytes, PPARγ activity is pivotal for glucose uptake via Glut4. It causes differentiation of preadipocytes and insulin sensitivity [13]. In two studies, no repressive effect of clozapine on PPARγ was detected [190,191]. PPARγ rather increases expression of SREBP1C with concomitant suppression of lipases [192]. This antilipolytic effect of clozapine enhances triglyceride concentration in adipocytes [57] as well as weight gain in male rats [26]. In contrast, strong activation of the AhR by PCB126 or alcohol represses PPARγ and fat synthesis in adipocytes, leading to wasting syndrome and weight loss with impairments in insulin/IGF-1/Akt signaling [193,194]. Moreover, adipose triglyceride lipase (ATGL)-driven lipolysis and extracellular release of these fatty acids leads to hyperlipidemia—another marker of MetS (Figure 1).

Vive versa, the uptake of fatty acids from the circulation, as well as de novo lipogenesis, triggers liver damage and metabolic dysfunction-associated steatotic liver disease (MASLD), which is considered to be the hepatic manifestation of MetS. All three chemicals provoke MASLD (Table 2). Hypertrophy, vacuolization, and glycogen depletion have similar findings during all treatments. In the liver and skeletal muscle, the PPAR isoform α controls lipid metabolism. It regulates genes involved in fatty acid oxidation pathways, fatty acid transport, and triglyceride catabolism (194). Ferno et al. [195] demonstrated a marked transient downregulation of the AhR target gene *pparα* at 6h after clozapine injection, which is relevant for the observed hepatic accumulation of fat in male rats. After downregulation of the AhR, PPARα activity increases and induces hepatomegaly in PCB126 and clozapine-treated male rats [196] (Table 2).

Similar to clozapine treatment, alcohol overconsumption induces insulin resistance because Akt phosphorylation is dramatically reduced in liver and adipose tissue [197]. Furthermore, AhR activation by alcohol decreases hepatic AMPK phosphorylation and autophagy and causes mitochondrial dysfunction [198]. Alcohol impedes hepcidin production and increases hepatic iron storage and ROS production, thereby worsening liver injury [198,199]. 

Last but not least, PCB126 causes insulin resistance, hepatic steatosis, and fatty-acid oxidation due to decreased expression of PPARα and its targets. The expression levels of enzymes involved in gluconeogenesis and glycogenolysis were strongly downregulated, thereby exacerbating hypoglycemia in fasting rats [193]. Despite this pseudo-starvation, the ligand-activated AhR inhibits AMPK activation, which is necessary for autophagy induction and recycling of cellular components [199]. 

Nonmedical treatment of steatosis includes abstinence from alcohol or protection from environmental pollutants. Gallage et al. [200] could ameliorate MASLD in mice under a Western diet with an intermittent fasting regime. Fasted mice had significantly lower levels of serum cholesterol and fasting blood glucose compared to mice under the same diet without intermittent fasting, although the benefits from fasting were independent of total calorie intake. Altered whole-body metabolism with increased fatty acid oxidation and ketogenesis, together with AMPK activation during the fasting cycles, seem to prevent lipid accumulation in the liver. 

For patients with treatment-resistant schizophrenia, clozapine is the last treatment option. The medication cannot be stopped, and additional drugs to treat MetS are necessary. Medical treatment of metabolic derangements uses insulin sensitizers or PPAR agonists like fenofibrate and rosiglitazone to restore liver functions [201,202,203]. Medication with PPAR agonists or the antidiabetic drug metformin activates AMPK, which uncouples the insulin/insulin receptor/Akt2/Glut4 pathway and triggers Glut4 translocation. Along this pathway, metformin and PPAR agonists ensure sufficient glucose uptake [69,204] and thereby improve liver lesions in rats with MASLD [205,206].

Moreover, this antidiabetic drug prevented inflammatory complications after COVID-19 infection [207] and was shown to reduce the risk of age-related diseases, including MetS, cardiovascular diseases, cancer, and neurodegeneration, even in non-diabetic patients [208]. Metformin is a safe, cheap, and effective drug for eliminating the energy deficits of all the diseases presented in Figure 1. However, further research is needed to examine possible negative side effects of metformin treatment more profoundly. Recently, Banitti et al. [209] described the positive effects of metformin on weight and cognition in patients under APD treatment, but in some patients, symptoms attributable to a relapse of schizophrenia were diagnosed. 

As shown in Table 2, female rats treated with clozapine are protected from hepatic damage probably by activation of the glycerol shunt, which dampens insulin secretion despite hyperglycemia [20,210]. Along this newly discovered pathway, glycogen/glucose-derived glycerol-3-phosphate is hydrolyzed to glycerol, which counters metabolic stress. Furthermore, the empty glycogen stores activate AMPK, which inhibits lipogenesis in the liver and induces autophagy. In contrast, the PCB-treated rats have insufficient AMPK activation despite glycogen depletion [193] (Table 2), but increased fatty acid oxidation in the liver points to some AMPK activity in the female PCB-treated rats, too. Thus, estrogen efficiently counteracts AhR activation in female rats.

Evaluation of all the experimental approaches allows for the conclusion that AMPK activation protects from steatosis and inflammation. Thus, AMPK-mediated glucose uptake or autophagy, respectively, seem to be attractive targets for age- and drug-related metabolic disorders like insulin resistance, weight gain, and hepatic steatosis [211]. 

## 5. Metabolic Changes in the Brain

Neuronal circuits within the hypothalamus play main roles in the regulation of whole-body metabolic homeostasis. They orchestrate physiological and behavioral responses that accommodate the energy and glycemic requirements of the body, including changes in food intake, energy expenditure, glucose production, and utilization. Interestingly, ESR1 and AhR expression are very high in hypothalamic neurons. Rats with MetS PI3K/Akt and AMPK signaling pathways in the hypothalamus are affected [212]. Similarly, clozapine and alcohol reduce insulin sensitivity and glucose uptake in the hypothalamus [194,213], which in turn induces cellular survival pathways.

### 5.1. The Brain–Liver Axis

Clozapine activates neurons in the medial prefrontal cortex with a wide range of projections to the mesolimbic, amygdala, and thalamic areas [214]. Yet it is not clear whether the newly discovered connection from the medial amygdala to the liver [215] or the hypothalamus–liver axis promotes the rapid synthesis of glucose by hepatic gluconeogenesis seen in clozapine-treated rats of both sexes (Figure 3). 

The APD-induced weight gain closely resembles ‘hypothalamic obesity’ [216], where the damage of hypothalamic nuclei often leads to decreased sympathetic activity, hyperphagia, rapid and excessive weight gain with central insulin and leptin resistance, decreased metabolic rate, and increased energy storage in adipose tissue. 

### 5.2. The Brain–Fat Axis 

Leptin is the key mediator of energy metabolism. It is produced primarily by adipocytes and communicates energy status to the brain via leptin receptors [217]. However, von Wilmsdorff et al. did not find specific differences in hypothalamic leptin receptor expression and leptin serum levels among APD-medicated Wistar rats [218]. In rats with hypothalamic obesity, chronic hyperleptinemia is linked to upregulation of the hypothalamic leptin receptor, while the adiponectin receptor 1 is downregulated [217]. Surprisingly, oral feeding of iron oxide nanoparticles corrected all deranged hormone levels and obesity within 4 weeks of treatment [217].

A negative correlation between obesity and circulating adiponectin is well established [219], and adiponectin concentrations increase concomitantly with weight loss, whereas decreased adiponectin levels are associated with insulin resistance and hyperinsulinemia [220]. Surprisingly, serum adiponectin levels are increased in all clozapine-medicated groups, indicating a (pseudo)-starvation-like status [26]. Higher adiponectin levels in the rats contrast with the decreased adiponectin secretion from 3T3 cells under clozapine treatment [221]. This contradiction between in vivo and in vitro studies points to the involvement of the brain–fat axis. As demonstrated in Figure 3, previous studies indicate neuronal energy deficits in the hypothalamus, which might be restored by adiponectin. Interestingly, Cisternas et al. [222] observed a restored glucose metabolism via AMPK activation and improved cognitive functions during adiponectin treatment in a MetS model for Alzheimer dementia [223]. In addition, hypothalamic AMPK induction by metformin suppresses oxidative stress and amyloidogenic processes in the MetS rat model [212]. 

### 5.3. The Gut–Brain Axis

The gut communicates with the brain via a bidirectional pathway called the gut–brain axis, which is involved in maintaining metabolic homeostasis. Gut microorganisms connect to the CNS through interconnecting pathways, including neurological, endocrine, and immunological signaling. Gut microbiota degrade proteins and amino acids. The degradation product of tryptophan, kynurenine, is an AhR ligand and has immune- and neuromodulating properties [224]. A disturbed gut–brain axis allows the bidirectional spread of proinflammatory molecules, which may contribute to the development/progression of metabolic diseases. 

Elevated kynurenine/ tryptophan ratios were detected in the blood and brain of schizophrenic patients [225], and experimental stroke induces the L-kyneurenine/AhR pathway, too [226]. However, in cell culture experiments, the AhR is activated through lysosomal cystine depletion [98] without the involvement of the brain and microbiota or their metabolites [111]. 

### 5.4. Activation of the Noradrenergic System under MetS

The activity of the serotonergic nervous system is suppressed by clozapine, while the noradrenergic system is activated [227]. Norepinephrine initiates lipolysis in adipocytes and favors the beiging of fat cells and thermogenesis [112,228]. Increased circulating hormone levels are involved in the alcohol- and PCB126-linked weight loss with hepatic triglyceride accumulation [229,230], suggesting that increased noradrenergic activity takes part in the development of hepatic steatosis. In line with this assumption, reduction in catecholamine synthesis and norepinephrine release prevents hepatic steatosis in mice. [231]. Furthermore, elevated epinephrine and norepinephrine levels result from a significant reduction in hepatic catechol-O-methyltransferase (COMT), which deactivates extra-neuronal norepinephrine [232]. *Comt* might be an AhR target gene because postnatal exposure to AhR agonists reduced hepatic COMT expression but increased it in the brain [233]. 

## 6. Clinical Interventions

In clozapine-treated patients with MetS, the outcome of lifestyle-changing interventions, including regular exercise and a healthy diet, is disappointing [234]. Some randomized-controlled trials support the use of metformin, aripiprazole, and Orlistat (in men only) for treating obesity [235]. Meta-analysis shows a robust effect of metformin in reducing body mass index and waist circumference but no effects on blood glucose, triglyceride levels, or HDL levels. Persisting hyperglycemia coincides with the high incidence of metformin-resistant clozapine-induced prediabetes/diabetes [236]. There is also very limited preclinical and clinical evidence that a combination of sodium-glucose cotransporter-2 (SGLT2) and metformin, in cases of DM2, has favorable metabolic effects [237]. In contrast, alternative comedication with glutathion precursor N-aceltylcysteine (NAC) showed effectivity in the treatment of schizophrenia, but unfortunately, metabolic changes were not investigated in this study [238]. In addition, a meta-analysis provided evidence that NAC improved all metabolic parameters in women with polycystic ovary syndrome [239]. The same positive metabolic effects are observed in rats with fructose-induced MetS [240].

Despite these promising results, further large-scale research is needed in the field of the management of metabolic dysfunctions in schizophrenic patients who undergo treatment with antipsychotic drugs.

## 7. Conclusions

The pathophysiology of MetS is incredibly complex. This review tries to elucidate the highly probable trigger and the network of signaling pathways that link the apparently unrelated diseases (Figure 1 and Figure 5). The trigger is insufficient glucose uptake due to insulin resistance, obesity, hypercortisolemia, or receptor-mediated Akt inhibition.

AhR is a convergence point of these signaling pathways that inform the cell of its external and internal environments. Cellular glucose deprivation under clozapine activates the AhR, which directs the sex-specific physiological changes in the liver/adipose-hypothalamic crosstalk [15,20,26]. AhR signaling reduces cellular iron availability at the expense of increased hepatic storage of excess iron [61,185]. Iron depletion stabilizes the transcription factor HIF-1, which increases the transcription of glucose transporters and VEGF. In addition, HIF-1 shifts metabolism toward glycolysis. Glycolytic ATP production is less effective than oxidative phosphorylation in the mitochondria and needs even higher amounts of glucose. Aberrant HIF-1 activity was detected in the peripheral blood cells of patients with MetS, T2DM, or obesity [241].

Patil et al. [242] denote the AhR as a ‘double-edged sword’. On the one hand, AhR triggers the HIF and ATF4 survival pathways; on the other hand, AhR activation blocks two alternative survival pathways.

The fuel-sensing enzyme AMPK dampens cellular energy consumption, activates recycling processes in the cell, and enhances glucose uptake by Glut4 translocation to the cell membrane. Active AhR triggers dephosphorylation and inactivation of AMPK (Figure 4).

Furthermore, AhR activity impairs cytoprotective H_2_S production [243] with its beneficial effects on glucose metabolism [244], inflammation, mitophagy [245], and vascular functions [246]. Low H_2_S levels are linked to MetS- and diabetes-associated complications [246,247]. On the other hand, H_2_S donors induce a hypometabolic state during illness, which limits organ injury. Refilling the cystine stores either directly with NAC or by upregulation of cysteinases and H_2_S production would dampen or even stop AhR activity. Together, they constitute a fascinating new therapeutic perspective for the treatment of MetS patients [248].

## Figures and Tables

**Figure 1 biomedicines-12-02294-f001:**
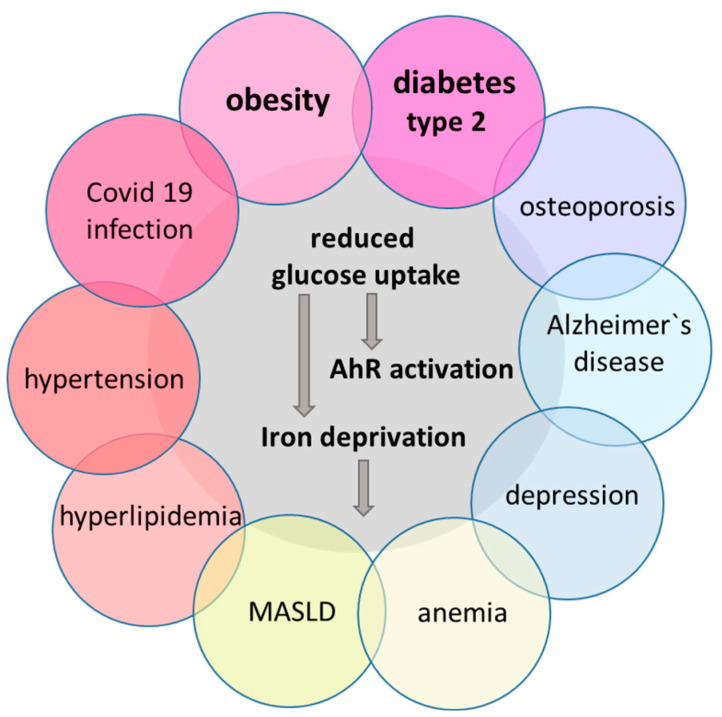
The interaction of MetS with steatosis, anemia, osteoporosis, and psychiatric diseases. Metabolic syndrome (MetS) is a cluster of risk factors—obesity, insulin resistance, dyslipidemia, elevated blood pressure, and hyperglycemia (reddish circles)—that increase the risk for cardiovascular disease, cognitive diseases, osteoporosis, and type 2 diabetes. In recent years, SARS-CoV2 infection was added as a potential risk factor for MetS. The mechanisms underlying MetS are complex and might include lipid accumulation, mitochondrial dysfunction, altered calcium homeostasis, and increased oxidative stress. This review provides evidence that reduced glucose uptake activates the AhR. This transcription factor induces genes that trigger iron depletion in order to stabilize the transcription factor hypoxia-induced factor 1 (HIF-1) [1,2]. In turn, HIF-1 induces transcription of glucose transporters to counteract glucose deprivation [1]. Unfortunately, the AhR also induces genes that are involved in lipogenesis with unfavorable effects like dyslipidemia [3,4]. Worsening of the metabolic parameters is linked to metabolic dysfunction-associated steatotic liver disease (MASLD) [3,4] and anemia [1] (green circles). Understanding the common molecular mechanisms (gray circle) unraveled in this review might help to break this vicious circle.

**Figure 2 biomedicines-12-02294-f002:**
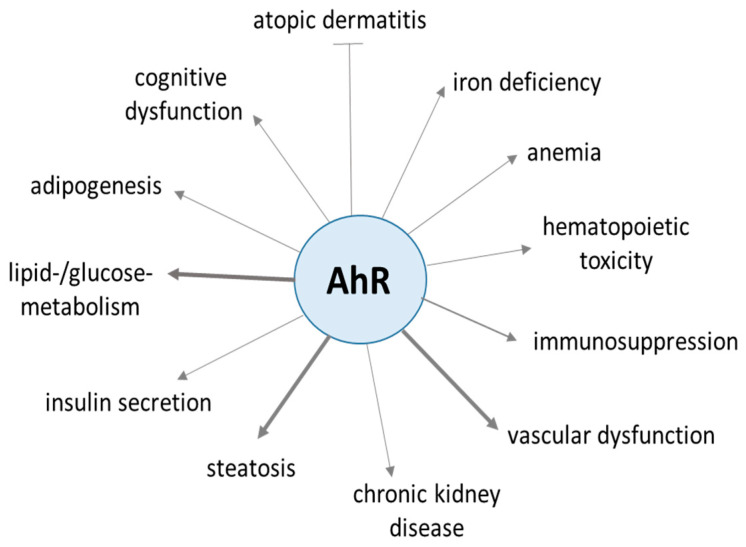
Overview of the (patho-)physiological effects linked to activation of the aryl hydrocarbon receptor (AhR). The widespread effects are observed in skin [5], bone marrow [6,7,8,9], endothelium [10], kidney [11], liver [12], pancreas [13,14], fat tissue [15] and even brain [16]. Thicker arrows mark the well-known striking effects of the AhR on glucose/lipid metabolism, liver, and vascularization. Further information on the role of the AhR in the different diseases is given in the cited literature.

**Figure 4 biomedicines-12-02294-f004:**
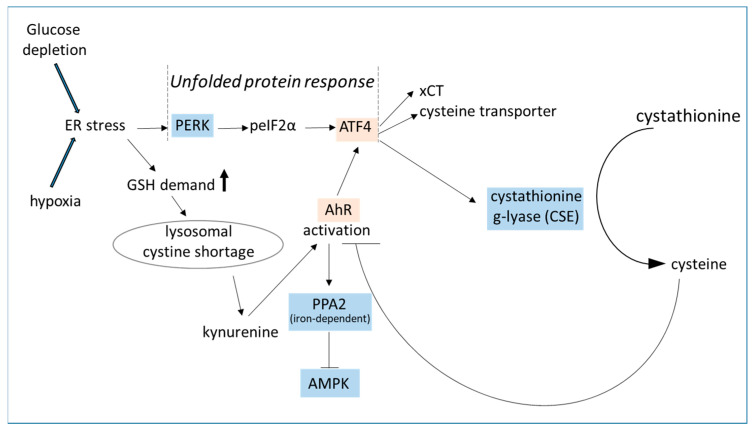
Non-canonical activation of the AhR. Either glucose depletion or hypoxia induces (oxidative) stress in the endoplasmatic reticulum (ER). The subsequent unfolded stress response along the PERK-eIF2α-ATF4 axis restores the intracellular GSH pool by increased uptake of cystine and cysteine and additional cysteine production by the cysteinase CSE. Furthermore, low cystine levels in endolysosomes trigger degradation of tryptophan to kynurenine, which is an endogenous AhR ligand [98]. Activation of the AhR induces the transcription of genes for ATF4 and PP2A [145,146]. PP2A is a serine/threonine dephosphorylating enzyme complex that inhibits the activation of AMPK. Very low iron availability inactivates PP2A, and AMPK can be activated by phosphorylation [101]. Transcription factors are highlighted in pink; cytosolic enzymes in blue. AhR—aryl hydrocarbon receptor; AMPK—AMP-activated protein kinase; ATF4—activating transcription factor; CSE—cystathionine-γ-lyase; eIF—eukaryotic initiation factor; GSH—glutathione; PERK—protein kinase R-like ER kinase; PP2A—protein phosphatase 2A; xCT—cystine/glutamate exchanger.

**Figure 5 biomedicines-12-02294-f005:**
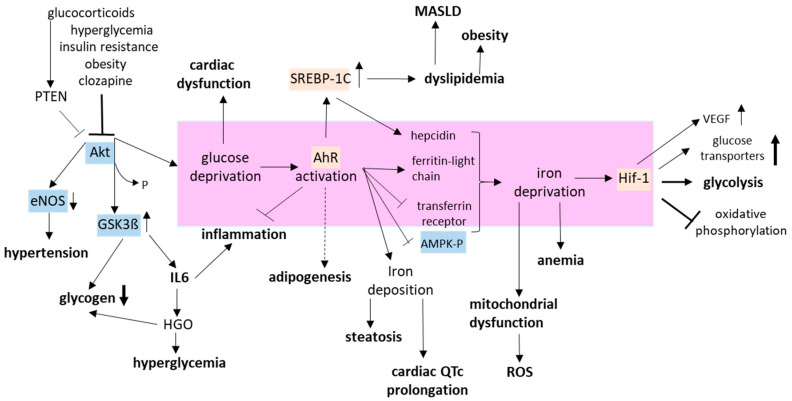
Interaction of biochemical pathways contributing to MetS. Aging, obesity, hyperglycemia, increased cortisol, insulin resistance, missing growth factors (GF), or blockade of neurotransmitter signaling by clozapine disrupts Akt signaling [69]. Missing Akt activation reduces nitric oxide production and endothelial-dependent dilatation [30,176]. In addition, active GSK3ß increases proinflammatory IL6 levels [125], which promote hepatic glucose output and hyperglycemia [156] to counteract the reduced cellular glucose uptake due to Akt inactivation. Under glucose deprivation, the cells secrete IL6 and other cytokines to request assistance from immune cells [106]. Intracellular glucose and subsequent glutathione deficiency activate the AhR [98]. The AhR reduces intracellular iron by ferritin expression and downregulation of the transferrin receptor CD71 [6], thereby increasing the risk for steatosis [185], cardiac QTc prolongation [182], anemia [1], and mitochondrial dysfunction [150]. Cellular iron deficiency stabilizes HIF-1α [103]. This transcription factor increases the expression of glucose transporters (GLUTs) and VEGF [102,106], which can, in turn, induce Akt phosphorylation [106]. Furthermore, AhR induces SREBP-1C expression, leading to increased lipogenesis in liver and fat tissue [26]. In contrast, the AhR inhibits the activation of the AMP-activated protein kinase (AMPK) by PP2A induction [146]. AMPK reduces energy consumption, starts autophagic recycling of cell substituents, and takes over Glut4 translocation and glucose uptake [79,80]. AMPK activation would favor fatty acid oxidation and oxidative phosphorylation (oxP) in the mitochondria, while HIF-1 shifts energy production toward glycolysis [103]. Transcription factors are highlighted in pink; cytosolic enzymes in blue. The direct role of the AhR in HIF-1 activation is presented in the pink box. AhR—aryl hydrocarbon receptor; AMPK—AMP-activated protein kinase; eNOS—endothelial nitric oxide synthase; GSK—glycogen synthase kinase; HGO—hepatic glucose output; IL6—interleukin 6; MASLD—metabolic dysfunction-associated steatotic liver disease; ROS—reactive oxygen species; SREBP—sterol regulatory element binding protein.

**Table 1 biomedicines-12-02294-t001:** The extent of glucose deprivation determines which survival pathway is activated. Moderate to severe cellular glucose depletion causes ER stress and triggers AhR activation. Subsequent ATF4 and HIF-1 activation provoke VEGF production, which can reactivate Akt signaling. Furthermore, ATF4 induces the transcription of genes that are involved in beneficial cysteine and H_2_S production. H_2_S, as well as massive energy loss or iron depletion, activate AMPK. This kinase intracellularly activates Akt2 signaling with high numbers of Glut4 on the cell membrane. Furthermore, Akt inhibition by corticosteroids, insulin resistance, or receptor blockade directly causes GSK3ß activation, which liberates the transcription factor NFκB. Via IL6 production, immune cells are attracted to and secrete VEGF. Therefore, the immune cells help to reconstitute glucose uptake. Arrows in Table 1 indicate the up- or downregulation of the respective proteins and pathways. AhR—aryl hydrocarbon receptor; ATF4—activating transcription factor 4; GFs—growth factors; Glut—glucose transporter; HIF-1—hypoxia-induced factor; H_2_S—hydrogen sulfide; AMPK—AMP-activated kinase; AMP—adenosine monophosphate; NFκB—nuclear factor kappa B; IL6—interleukin 6; VEGF—vascular endothelial growth factor.

Glucose Uptake	Mediators	Trigger	Pathways to Reconstitute Glucose Uptake		References
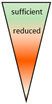	Akt	GFs, hormones, neurotransmitter	translocation of Gluts into the membrane	glucose uptake↑	[69]
	AhR	Cystine depletion→kynurenine (?)	HIF-1 and ATF4	glucose uptake↑	[98]
	ATF4	ER/mitochondrial stress	H_2_S production → AMPK-P, Glut5↑	glucose uptake↑	[99,100]
			VEGF expression ↑ → Akt-P	glucose uptake↑	[101,102]
	H1F1	iron deprivation	expression of Gluts		[103]
	AMPK	High AMP, iron deprivation, H_2_S	Glut4 translocation ↑, Glut4 recycling ↓	glucose uptake↑	[80]
Akt inhibition	GSK3β	AhR activation			[104]
		NFkB → IL6	help of immune cells → VEGF ↑	glucose uptake↑	[105,106]

**Table 2 biomedicines-12-02294-t002:** Shows a tabular comparison of the metabolic effects of AhR activation in the animal model. The different AhR inducers were the strong AhR ligand PCB126, clozapine, and EtOH. The first two studies were conducted with male and female Sprague–Dawley rats, and the EtOH study was with male rats only. One dose of PCB126 was injected i.p., and the animals were killed after 26 days. Clozapine was given together with the food for 12 weeks. EtOH was given intragastrically for 4 weeks. All treatments provoke liver damage and metabolic disturbances in glucose and lipid metabolism. Arrows indicate up- or downregulation of the listed physiological parameters. Question marks represent missing information in the corresponding study. AhR—aryl hydrocarbon receptor; EtOH—ethanol; MDA—malondialdehyde; pAMPK—phosphorylated AMP-activated protein kinase; SREBP—sterol regulatory element binding protein; TGs—triglycerides; wt—wildtype.

		AhR wt Female	AhR wt Male	AhR wt Female	AhR wt Male	AhR wt Male
AhR Induction by	PCB126	Clozapine	EtOH
blood	weight	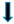	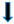		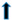	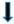
blood glucose	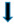	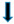	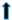		?
Cholesterol	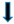		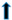		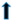
TGs	?	?		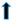	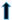
liver	weight	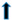	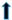		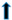	?
vacuolation	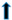	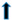		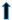	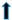
hypertrophy	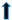	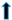	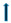	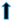	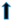
glycogen	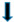	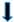	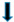		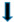
gluconeogenesis	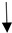	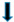	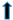		
fatty acid oxidation	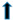		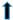		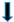
pAMPK	?	?	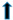		
SREBP-1C	?	?		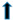	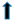
MDA	?	?		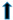	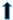

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
