# Peer review of "Metabolic Side Effects from Antipsychotic Treatment with Clozapine Linked to Aryl Hydrocarbon Receptor (AhR) Activation"

_biomedicines, 2024, doi:10.3390/biomedicines12102294_

Round 1

Reviewer 1 Report

Comments and Suggestions for Authors

"  

This study explores the mechanism by which clozapine contributes to metabolic reprogramming. That is the effect of the drug on cell metabolism and cell signaling pathways, of which AhR activation is one. Clozapine causes transient inhibition of Akt and glucose uptake, which leads to ER stress and altered mitochondrial respiratory functions, thus affecting metabolic syndrome. This knowledge can lead to the design of targeted therapy for the management of metabolic side effects in patients being treated for psychosis with antipsychotics, such as clozapine.

1.      Some sentences are long and complex and can be simplified to improve readability. Simplifying complex sentences to improve readability. For example, "Due to the halogenation of the clozapine molecule and the similar metabolic effects, AhR activation by clozapine was supposed. “Although a plethora of molecules and signaling pathways associate with MetS, our understanding of how these pathways interconnect to coordinate organismal physiology remains limited” is too complex. Breaking it into two sentences improves the clarity. "Activation of the HIF1/VEGF pathway shifts the metabolism of HL60 cells towards glycolysis and slows down the Krebs cycle." Simplify for clarity and readability, another example "Neuroreceptor inhibition under clozapine transiently reduces Akt activation and glucose uptake, which induces ER stress and the unfolded protein response (UPR)."

2.      For some points, such as AhR activation by glucose for glucose deprivation, more direct mechanisms or examples might have been better extrapolated to further strengthen the argument.

3.      The paper would benefit from a somewhat more structured approach, sharply separating sections with subtitles introducing each major topic, such as environmental pollutants, AhR activation, and drug-induced metabolic effects.

4.      Some terms and processes, such as "xenobiotic metabolism" and "heterodimeric, The term "mitohormesis" is used without prior explanation. complex," could be briefly defined or explained for readers who may not be familiar with them

5.       This argument would be credible by offering information on specific examples or data from studies on the link between the serum concentrations of pollutants and MetS.

6.      On the other hand, the statement "One can assume that AhR is triggering the other metabolic side effects of clozapine as well" could have been substantiated with some examples of specific evidence or references.

7.      More examples or data on the specific ways AhR activation leads to sex-specific physiological changes might be provided.

8.      Evidence or reference may support assertions, such as the role of AhR in clozapine's effects

9.       Elaborating the broader implications of AhR activation in different tissues.

10.   A detailed description of the figure is provided in the text, explaining each component of the figure and its importance to the study.

11.  Figure 2 Keep symbols and colors consistent for the same parameters across figures for readability.""

Reviewer 2 Report

Comments and Suggestions for Authors

   1. The abstract should be more informative and cover Background, Methods, Results, and Conclusion.

  2. The author needs to summarize the key findings in the abstract for a quick understanding.

3. The introduction provides a good overview but could be more concise. Emphasize the significance of the study and the main research question.

  4. The author needs to add a brief outline of the paper's structure at the end of the introduction to guide readers.

5. The author needs to clearly state the gaps in the existing literature that this study aims to fill.

6. The author needs to explain the rationale behind the choice of animal models and how they relate to human metabolic responses.

7. The author needs to present the data in a more structured manner, using subheadings for different sections (e.g., metabolic changes, sex differences).

8. The author needs to ensure that all figures are clearly labeled and include legends.

9. The author needs to highlight the implications of the findings for clinical practice and future research. Discuss the limitations and potential biases.

10. The conclusion should briefly recap the main findings and their significance. The author need to avoid introducing new information.

11. The author needs to check for consistency in the use of abbreviations and terms throughout the document. Provide a list of abbreviations if necessary. Ensure that all technical terms are explained when first introduced.

12. The author needs to ensure all figures and tables are high quality and adequately described in the text.

13. The authors need to label all axes and include units of measurement where applicable.

14. Figures 1 and 2: These figures are critical in understanding the pathways involved. Ensure they are high-resolution and clearly annotated.

15. The discussion on sex differences is intriguing.  The author needs to ensure this section is supported with adequate data and references.

16. The review discusses various pathways (e.g., AhR activation, AMPK activation) in detail. Consider summarizing these pathways in a table for clarity.

17. Some references appear older and might need to be complemented with more recent studies. Consider adding more recent studies to support the findings and discussion points

18. Proofread the document for grammatical errors and improve the flow of sentences.

Comments on the Quality of English Language

Moderate English revision is required 

Reviewer 3 Report

Comments and Suggestions for Authors

The authors presented a review on metabolic side effects under antipsychotic treatment linked to aryl hydrocarbon receptor (AhR) activation. I think the manuscript is informative for researchers as well as clinicians, however the authors should revise it according to the following concerns;

1.      The authors should describe more in detail on the clinical significance of metabolic side effects due to antipsychotics found in psychiatric disorders for the treatment of which antipsychotic treatment is applied, citing relevant literatures.

2.      The authors mainly focused on metabolic side effects and their mechanism due to clozapine. The authors should also describe metabolic side effects and their mechanism of other antipsychotics. If there are any differences in metabolic side effects and their mechanism between clozapine and other antipsychotics, the authors should discuss whether or not they are related to unique clinical profiles of clozapine.

Comments on the Quality of English Language

fair

Reviewer 4 Report

Comments and Suggestions for Authors

Karin Fehsel's review is entitled „Metabolic side effects under antipsychotic treatment are linked to aryl hydrocarbon receptor (AhR) activation.” Based on the title, the review is expected to provide details about the induction of metabolic syndrome (MetS) by antipsychotic drugs. Abstract is very encouraging with olanzapine and clozapine and differences in the occurrence of symptoms (MetS) depending on gender in rats. The manuscript is divided into 1. Introduction and a few subchapters ending with conclusion. Three figures additionally fulfill the manuscript.

While the manuscript holds significant potential for the scientific community, it is crucial to acknowledge that it requires further refinement before publication (Reconsider after major revision).

General comments:

1.      At present form, the manuscript seems to be a draft for a good review article

While the manuscript offers a compilation of information on MetS, it requires more focus on the composition from general to detail to access the whole picture of causes and consequences of glucose deprivation separately in the brain and the rest of the human body.

2.      The title “antipsychotic treatment “suggests that the whole group of antipsychotics will be presented; however, inside the manuscript, sporadically, there is only clozapine. Why only?

The above is the reason why the title does not correspond with the text of the manuscript—the author probably did not have a concept regarding the layout/composition of the whole text.

3.      Lack of a clear concept of narration – starting from the whole idea to nitty-gritty elements of the manuscript. Such an approach results in a huge mess regarding conceptual and compositional aspects of the work. In significant part, the manuscript consists of sentences selected from various publications. Single sentences make sense, but their composition into a coherent whole is, in my opinion, wrong.

4.      Next, are the three figures in the draft version? The figures are not easy to interpret and understand. They poorly illustrate the text, and the captions are unclear (there are sentences taken from the text or vs).

5.      Last, the conclusion is very confusing. The reviewer has no idea why there is a paragraph regarding metformin in the conclusion.

Round 2

Reviewer 2 Report

Comments and Suggestions for Authors

1. The connection between clozapine-induced metabolic changes and aryl hydrocarbon receptor (AhR) activation is central to the paper. However, the hypothesis linking AhR activation to specific metabolic disturbances could be clarified in the introduction. How directly are AhR's effects attributed to clozapine versus other pathways?

2. There are instances where terminology such as "MetS" (Metabolic Syndrome) and "AhR activation" could be more consistently applied or defined earlier for non-specialist readers. The author needs to ensure that all abbreviations are consistently used across sections.

3. The review integrates complex pathways like Akt, AMPK, and HIF-1α. However, more visual aids like diagrams could significantly enhance understanding, particularly in sections describing molecular mechanisms.

4. The introduction states that the diseases related to metabolic syndrome (MetS) are interconnected but doesn't delve into why clozapine's impact is particularly pronounced in metabolic pathways. Could this be explored in more detail?

5. The review notes a relationship between antipsychotic drugs and MetS, but the rationale for choosing clozapine (and not other antipsychotics) could be reinforced.

6. The relationship between AhR and glucose deprivation needs more detail to show how clozapine’s effects differ from other AhR ligands. Could the author expand on any novel findings related to AhR?

7. There’s mention of estrogen’s protective role in MetS, especially in female rats. Given that these findings may be age-dependent (as noted in the review), could this be investigated further in a human clinical context? For example, how would age and menopausal status in female patients influence clozapine-induced metabolic disorders? Were there any gender differences in human studies that corroborate the findings regarding estrogen’s protective role in female rats?

8. Given that mitochondrial dysfunction is highlighted, could clozapine’s impact on mitochondrial biogenesis and respiration be further explored?

9. Figures like Fig. 1 and Fig. 5 are helpful but should be explicitly referenced and explained in the results or discussion sections. Consider providing more detailed captions to ensure readers understand the implications.

10. A more detailed visual representation showing the interaction between AhR activation, AMPK, Akt, and HIF-1 pathways in the context of clozapine could improve the reader’s understanding.

11. The conclusion mentions that AhR is a "double-edged sword." This could be explored more explicitly by discussing the clinical implications for treating schizophrenia patients. Is there a therapeutic window where AhR’s effects are beneficial without causing significant metabolic disruption?

12. While the review extensively discusses molecular pathways, the translation to clinical practice could be stronger. Are there any recommendations for clinicians based on these findings? For instance, how should clinicians monitor metabolic changes in patients taking clozapine?

13. The author needs to ensure that all references, especially recent ones, are relevant and up-to-date. For example, the impact of COVID-19 on metabolic syndrome could be better integrated with references to the most current research.

 14. Would alternative antipsychotic drugs that do not activate AhR to the same extent provide a safer metabolic profile for schizophrenic patients?

Comments on the Quality of English Language

There are no major concerns regarding the English language in terms of grammar or overall readability. However, certain sections would benefit from clearer phrasing and enhanced articulation to improve the flow and precision of the ideas

Reviewer 3 Report

Comments and Suggestions for Authors

I think the manuscript has properly been revised according to the reviewers' comments.

Comments on the Quality of English Language

fair

Author Response

Thanks for the revision!

Reviewer 4 Report

Comments and Suggestions for Authors

Thank you very much for responding to the suggestions.

Author Response

Thanks for your revision!

Round 3

Reviewer 2 Report

Comments and Suggestions for Authors

The author has addressed the comments approperiatly 

Comments on the Quality of English Language

Minor editing of English language required.